# Nematic fluctuations in the cuprate superconductor $Bi_2Sr_2CaCu_2O_{8+\delta}$

N. Auvray [1], B. Loret[1], S. Benhabib[1], M. Cazayous[1], R.D. Zhong [2], J. Schneeloch[2], G.D. Gu[2], A. Forget[3], D. Colson[3], I. Paul[1], A. Sacuto [1] & Y. Gallais [1]*

Establishing the presence and the nature of a quantum critical point in their phase diagram is a central enigma of the high-temperature superconducting cuprates. It could explain their pseudogap and strange metal phases, and ultimately their high superconducting temperatures. Yet, while solid evidences exist in several unconventional superconductors of ubiquitous critical fluctuations associated to a quantum critical point, in the cuprates they remain undetected until now. Here using symmetry-resolved electronic Raman scattering in the cuprate $Bi_2Sr_2CaCu_2O_{8+\delta}$, we report the observation of enhanced electronic nematic fluctuations near the endpoint of the pseudogap phase. While our data hint at the possible presence of an incipient nematic quantum critical point, the doping dependence of the nematic fluctuations deviates significantly from a canonical quantum critical scenario. The observed nematic instability rather appears to be tied to the presence of a van Hove singularity in the band structure.

[1] Université de Paris, Matériaux et Phénomènes Quantiques, CNRS UMR 7162, F-75205 Paris, France. [2] Condensed Matter Physics and Materials Science Department, Brookhaven National Laboratory, Upton, NY 11973, USA. [3] Service de Physique de lÉtat Condensé, DRF/IRAMIS/SPEC (UMR 3680 CNRS), CEA Saclay, 91191 Gif-sur-Yvette cedex, France. *email: yann.gallais@univ-paris-diderot.fr

Unconventional superconductivity (SC) is often linked to the proximity of an electronically ordered phase whose termination at a quantum critical point (QCP) is located inside a superconducting dome-like region[1,2]. This observation suggests quantum criticality as an organizing principle of their phase diagram. The associated critical fluctuations could act as possible source for enhanced superconducting pairing[3–5] and explain their ubiquitous strange metal phases which often show non-Fermi liquid behavior, like a linear in temperature resistivity[7]. However whether the quantum critical point scenario holds for high-$T_c$ superconducting cuprates has remained largely unsettled[6–8]. This can be traced back to two fundamental reasons. First the exact nature of the pseudogap (PG) state, the most natural candidate for the ordered phase, remains mysterious. Experimentally a wealth of broken symmetry phases have been reported at, or below, the somewhat the loosely defined PG temperature $T^*$[9]. It is currently unclear which, if any, of these orders is the main driver of the PG phase as they could be all secondary instabilities of a pre-existing PG state. Second there is little direct evidence for critical fluctuations above $T^*$, questioning the very existence of a QCP associated to the termination of a second order phase transition at a critical doping $p^*$ (see Fig. 1a). AF fluctuations do not appear to be critical close to the putative QCP, at least for hole-doped cuprates[10,11], and critical CDW fluctuations are only observed below $T^*$[12]. Fluctuations associated to the more subtle intra-unit-cell orders[13–17] are more elusive and have not been reported up to now.

Recently nematicity, an electronic state with broken rotational symmetry but preserved translational invariance of the underlying lattice[18] (see Fig. 1b), has emerged as a potential candidate for the origin of the PG phase. A second order phase transition to a nematic phase, breaking the $C_4$ rotational symmetry of the $CuO_2$ plane, has been reported by torque magnetometry in $YBa_2Cu_3O_{6+\delta}$ close to the $TT^*$ reported by other techniques[19]. In a separate study a divergent electronic specific heat coefficient was observed at the endpoint of the PG phase in several cuprates and was interpreted as a thermodynamical hallmark of a QCP[20]. The nature of the ordered state associated to this putative QCP is however not yet settled, and nematicity stands as a potential candidate. To assess its relevance and the role of nematic degrees of freedom in driving the PG order, probing the associated fluctuations is thus essential.

Because it probes uniform ($q = 0$) dynamical electronic fluctuations in a symmetry selective way, electronic Raman scattering can access nematic fluctuations without applying any strain even in nominally tetragonal systems[21]. For metallic systems, the nematic fluctuations probed by Raman scattering can be thought as dynamical Fermi surface deformations which break the lattice point group symmetry. In the context of iron-based superconductors (Fe SC) ubiquitous critical nematic fluctuations were observed by Raman scattering in several compounds[22–24]. They were shown to drive the $C_4$ symmetry breaking structural transition from the tetragonal to the orthorhombic lattice, and to persist over a significant portion of their phase diagram[21]. In the context of cuprates nematicity along Cu-O-Cu bonds has been reported via several techniques, mostly in $YBa_2Cu_3O_{6+\delta}$[14–17,19,25,26]. The associated order parameter is an uniform traceless tensor of $B_{1g}$ (or $x^2 - y^2$) symmetry, which switches signs upon interchanging the $x$ and $y$ axis of the $CuO_2$ square plane (Fig. 1b). Nematicity along different directions has also been found in one layer mercury-based cuprate $HgBa_2CuO_4$[27], and overdoped $La_{1-x}Sr_xCuO_4$[28]. In the former case nematicity develops along the diagonal of the $CuO_2$ plane and thus transforms as the $B_{2g}$ (or $xy$) symmetry. The ability of Raman scattering to resolve the symmetry of the associated order parameter is therefore crucial.

## Results

**Doping and symmetry dependent Raman spectra.** We present Raman scattering measurements on 6 single crystals of the cuprate $Bi_2Ca_2SrCu_2O_{8+\delta}$ (Bi2212) covering a doping range between $p = 0.12$ and $p = 0.23$. A particular emphasis was put in the doping region bracketing $p^* \sim 0.22$ close to where the PG was shown to terminate, i.e., between $p = 0.20$ and $p = 0.23$[29–36]. At these dopings a relatively wide temperature range is accessible above both $T^*$ and $T_c$ to probe these fluctuations, and look for fingerprints of a nematic QCP. The polarization resolved Raman experiments were performed in several configurations of in-plane incoming and outgoing photon polarizations in order to extract the relevant irreducible representations, or symmetries, of the $D_{4h}$ group: $B_{1g}$ which transforms as $x^2 - y^2$, $B_{2g}$ ($xy$) and $A_{1g}$. As indicated above while the former two correspond to nematic orders along and at 45 degrees of the Cu–O–Cu bonds, respectively, the latter one is fully symmetric and is not associated to any symmetry breaking. The recorded spectra were corrected by the Bose factor and are thus proportional to the imaginary part of the frequency dependent Raman response function $\chi''_\mu(\omega)$ in the corresponding symmetry $\mu$ where $\mu = B_{1g}, B_{2g}, A_{1g}$ (see Methods section for more details on the Raman scattering set-up and polarization configurations).

In Fig. 1c is displayed the evolution of the normal state Raman spectrum in the $B_{1g}$ symmetry as a function of doping. From previous Raman studies, OD60 ($T_c = 60K$) sample lies very close to the termination point of the PG, $p^* \sim 0.22$ and no signature of PG is seen for OD60, OD55 ($T_c = 55K$), and OD52 ($T_c = 52K$) compositions[36]. Other samples, OD74 ($T_c = 74K$), OD80 ($T_c = 80K$), and UD85 ($T_c = 85K$) display PG behavior. The normal state spectra are consistent with previously published Raman data for the doping compositions where they overlap[37]. In particular while the spectrum shows little temperature dependence in the underdoped composition UD85, it acquires a significant temperature dependence in the overdoped regime where the overall $B_{1g}$ Raman response increases upon cooling. The low-frequency slope of the Raman response being proportional to the lifetime of the quasiparticles, this evolution was previously attributed to an increase metallicity of anti-nodal quasiparticles, located at ($\pi$, 0) and equivalent points of the Brillouin zone with overdoping[37]. However the increase of intensity upon cooling observed in overdoped compositions, $p > 0.2$, is not confined to low frequencies as expected in a naive Drude model, but extends over wide energy range up to at least 500 cm$^{-1}$. This suggests that it is not a simple quasiparticle lifetime effect but, as we show immediately below, it is rather associated to a strongly temperature dependent static nematic susceptibility. It is also evident in Fig. 1c that this evolution is non-monotonic with doping as the OD60 spectra shows significantly more temperature dependence than at any other dopings.

**Symmetry-resolved susceptibilities.** To analyze the observed temperature dependence and its link to a nematic instability, it is useful to extract the symmetry resolved static susceptibility $\chi_\mu(\omega = 0) = \chi^0_\mu$ from the measured finite frequency response $\chi''_\mu(\omega)$ using Kramers-Kronig relations:

$$\chi^0_\mu = \int_0^\Lambda d\omega \frac{\chi''(\omega)}{\omega} \qquad (1)$$

where $\mu$ stands for the symmetry and $\Lambda$ a high-energy cut-off. In order to perform the integration, the spectra were interpolated to zero frequency either linearly, or using a Drude lineshape (see Supplementary Note 2 and Supplementary Figs. 2 and 3). The integration was performed up to $\Lambda = 800$ cm$^{-1}$, above

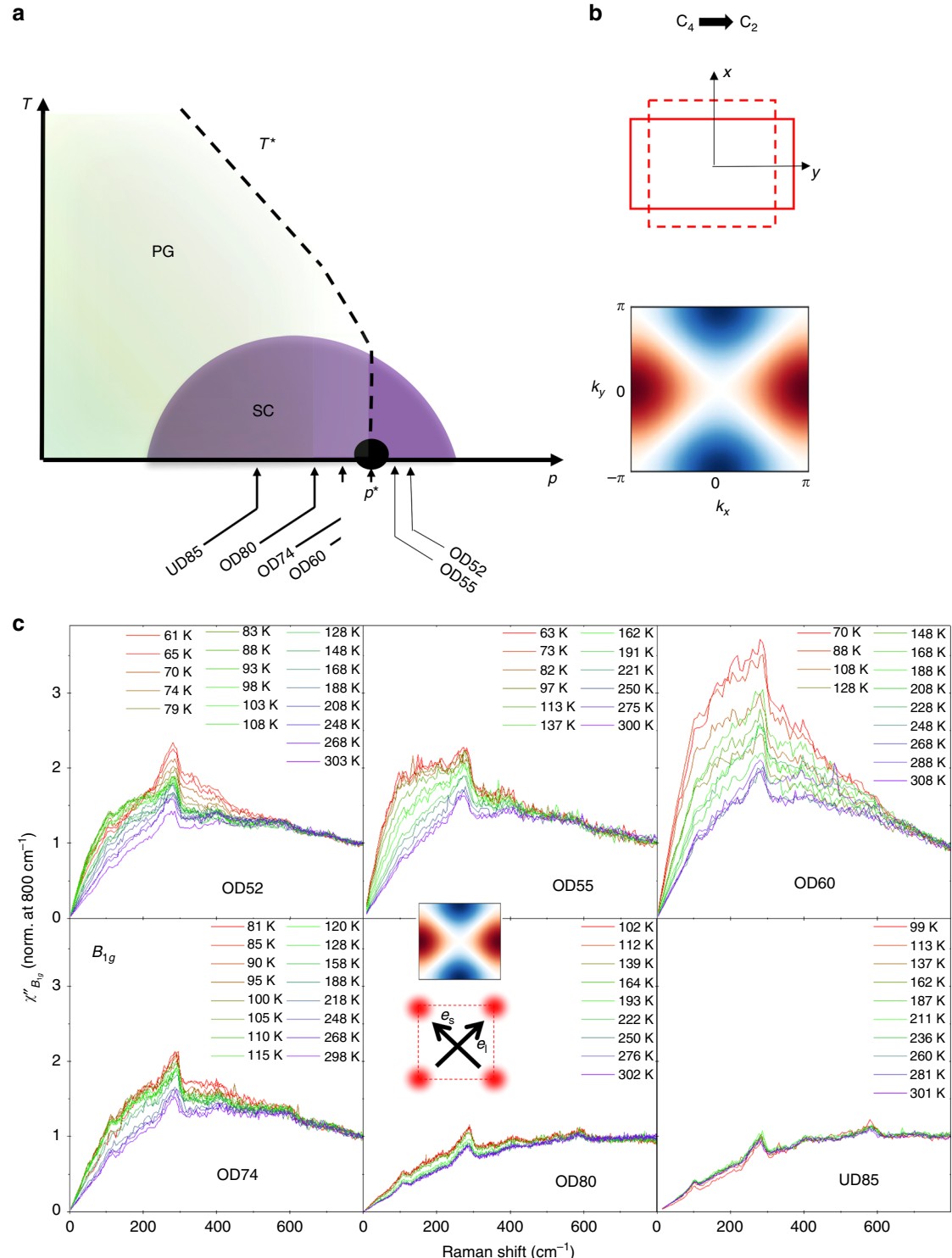

**Fig. 1** Dynamical nematic fluctuations in $Bi_2Sr_2CaCu_2O_{8+\delta}$ **a** Temperature-doping generic phase diagram of hole-doped cuprates. The pseudogap phase ends at a putative quantum critical point (QCP) located at the doping $p^*$. **b** Nematic order breaking the $C_4$ rotational symmetry of the Cu square lattice down to $C_2$ symmetry. The corresponding order parameter has $B_{1g}$ symmetry: in reciprocal space it transforms as $k_x^2 - k_y^2$ and switches sign upon 90 degrees rotation $x \rightarrow y$ (color scale is defined as blue: negative values, red: positive values and white: 0). **c** Temperature dependence of the $B_{1g}$ Raman spectrum in the normal state for several doping levels in $Bi_2Sr_2CaCu_2O_{8+\delta}$. The $B_{1g}$ symmetry is obtained using cross-photon polarizations at 45 degrees of the Cu–O–Cu direction (see insets)

which the spectra do not show any appreciable temperature dependence in the normal state (see Supplementary Fig. 4). For $B_{1g}$ symmetry $\chi^0_{B_{1g}}$ is directly proportional to the static electronic nematic susceptibility, and its evolution as a function of doping

and temperature is shown in Fig. 2a. For a comparison the same quantity, extracted for 3 crystals in the complementary symmetries (see Supplementary Note 1 for the spectra), $B_{2g}$ and $A_{1g}$, is also shown. In order to compare different compositions and since

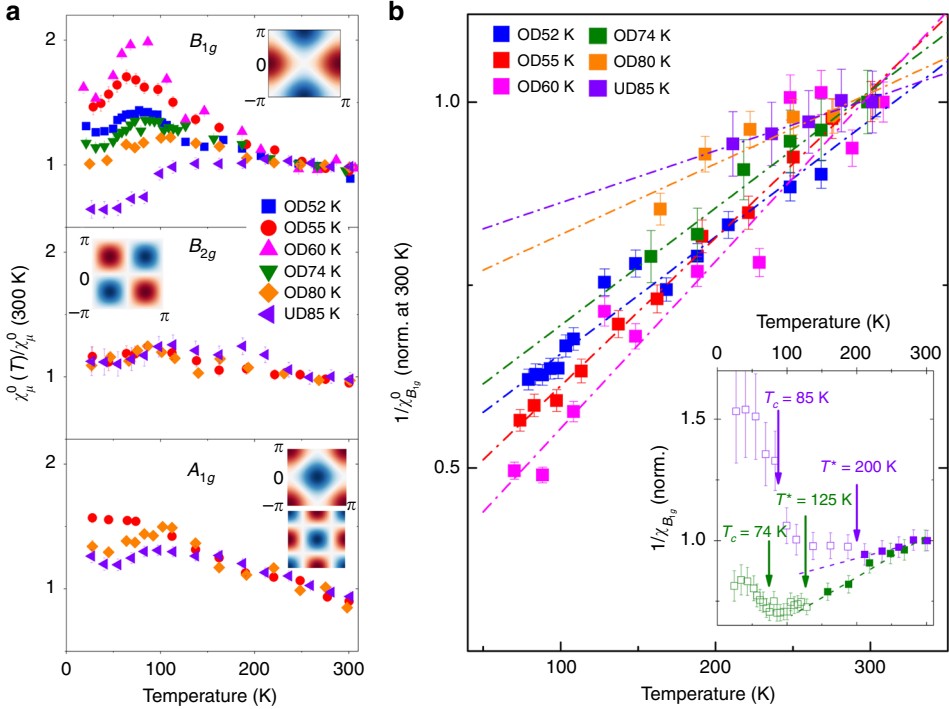

**Fig. 2** Symmetry resolved static susceptibilities. **a** Temperature dependences of the static susceptibility in 3 different symmetries, $B_{1g}$, $B_{2g}$, and $A_{1g}$, as a function of doping. The form factors for each symmetry are depicted in reciprocal space in insets. They are given in terms of the lowest order Brillouin zone harmonics: $\cos(k_x) - \cos(k_y)$ for $B_{1g}$, $\sin(k_x)\sin(k_y)$ for $B_{2g}$, $\cos(k_x) + \cos(k_y)$ and $\cos(k_x)\cos(k_y)$ for $A_{1g}$. The error bars correspond to the standard error of the low energy fits used for the low energy extrapolation (see supplementary note 1). **b** Curie-Weiss fits of the inverse $B_{1g}$ nematic susceptibility for temperatures above $\max(T_c, T^*)$. The inset shows the full temperature dependence of the inverse susceptibility of OD74 and UD85 where deviation from Curie-Weiss law are observed at $T^*$, and an additional upturn is observed at $T_c$. Full and open symbol correspond to data above and below $T^*$, respectively

we do not have access to the susceptibilities in absolute units, all extracted susceptibilities have been normalized to their average value close to 300 K, and we focus on their temperature dependences. The temperature dependence of the $B_{1g}$ nematic susceptibility is strongly doping dependent. In the normal state it is only weakly temperature dependent for UD85 ($p = 0.13$), while for overdoped compositions it displays a significant enhancement upon cooling. The pronounced divergent-like behavior for OD60 is only cut-off by the entrance to the SC state. This effect is however reduced for $p > p^*$ (OD55 and OD52), mirroring the non-monotonic behavior already apparent in the raw spectra. While our focus here is on the normal state, it is notable that the nematic susceptibility is suppressed upon entering the superconducting state for all doping studied, suggesting that the nematic instability is suppressed by the emergence of the superconducting order. In addition a weak but distinct suppression of $\chi_\mu^0$ is also observed above $T_c$ for UD85 ($\sim$200 K), close to the value of $T^*$ determined by other techniques in Bi2212 for similar doping levels[33,38]. By contrast the static susceptibilities extracted in $B_{2g}$ and $A_{1g}$ symmetries, while displaying some mild enhancement, are essentially doping independent above $T_c$. This symmetry selective behavior unambiguously demonstrates the nematic nature of the critical fluctuations observed close to $p^* \sim 0.22$.

**Curie-Weiss analysis of the $B_{1g}$ nematic susceptibility.** Further insight into the doping dependence of these critical nematic fluctuations can be gained by fitting the $B_{1g}$ static nematic susceptibility using a Curie-Weiss law:

$$\frac{1}{\chi_{B_{1g}}^0} = A \times (T - T_0) \qquad (2)$$

In a mean-field picture of the electronic nematic transition, such a behavior is expected on the high temperature tetragonal side of a second order phase transition which would set in at $T_0$. A negative $T_0$ implies that the ground state is on the symmetry unbroken side of the phase transition. Fig. 2(b) shows linear fits of the inverse susceptibility for all doping studied. Since clear deviations to linear behavior for $\frac{1}{\chi_{B_{1g}}^0}$ are seen below $T^*$ and $T_c$ (see inset of Fig. 2b), we restrict our fits to temperatures above $T^*$ for doping levels below $p^*$, and above $T_c$ for doping level above $p^*$. The fits allows us to extract $T_0$, the mean-field nematic transition temperature, which quantifies the strength of the nematic instability: graphically $T_0$ corresponds to the zero temperature intercept of the inverse susceptibility.

The evolution of $T_0$ as a function of doping is summarized in Fig. 3 in a phase diagram showing the corresponding evolution of the nematic susceptibility in a color-coded plot. Coming from the strongly overdoped regime, $p \sim 0.23$, $T_0$ increases towards $p^* \sim 0.22$ but upon further reducing doping, instead of crossing-over to positive temperatures $T_0$ reverses its behavior and decreases strongly, suggesting a significant weakening of the nematic instability below $p^*$. While the $T_0$ values remain negative at all doping suggesting the absence of true nematic quantum criticality, two aspects should be borne in mind. First the three $T_0$ values above $p^*$ extrapolate to $T = 0$ K at a doping level above the one corresponding to OD74, indicating the possible presence of a nematic QCP located between the doping levels corresponding to OD74 and OD60 crystals. Second, as shown in the context of Fe SC, the extracted susceptibility from Raman measurements does not include the contribution of the electron-lattice coupling[21]. In particular the linear nemato-elastic coupling is expected to increase the nematic transition temperature above $T_0$ and shift accordingly the location of the QCP[39]. This lattice-induced shift

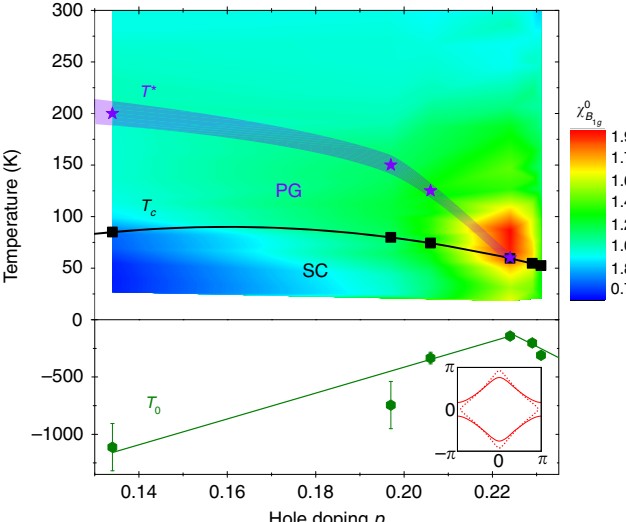

**Fig. 3** Phase diagram of critical nematic fluctuations. Color-coded plot summarizing the evolution of the $B_{1g}$ nematic susceptibility as a function of doping and temperature in Bi2212. The nematic Curie-Weiss temperature $T_0$ is also shown along with the superconducting $T_c$ and pseudogap $T^*$ temperatures. The lines are guide to the eye. The error bars for $T_0$ correspond to the standard error of the Curie-Weiss fits. The inset shows the Fermi surface deformation associated to the incipient Pomeranchuk instability which breaks the $C_4$ symmetry

## Discussion

We now discuss the possible origin of the observed nematic fluctuations, and then elaborate on the implications of our findings for the phase diagram of the cuprates. Theoretically two routes for nematicity have been proposed in the context of cuprates. The first route is via the melting of an uni-axial density wave order, like stripe or charge density wave, and is expected to apply to the underdoped regime where the tendency towards these orders is strongest[42]. It is unlikely to be relevant to our findings since the nematic susceptibility is almost featureless above $T^*$ in the underdoped crystal, and only shows significant enhancement in the overdoped regime. A second route is via a Pomeranchuk instability of the Fermi liquid, where the Fermi surface spontaneously deforms and breaks the underlying lattice rotational symmetry (see inset of Fig. 3). Theoretically, such an instability was indeed shown to be relevant to the cuprates close to doping levels where the density of states passes through a van Hove singularity (vHs) at the $(\pi, 0)$ and equivalent points of the Brillouin zone[43–45]. This is consistent with our results since $p^* = 0.22$ corresponds to the doping level where one of the Fermi surface sheets of Bi2212 changes from electron-like to hole-like, and passes through a van Hove singularity (vHs) at $(\pi, 0)$[29,36]. The link between the nematic instability and the closeness of the vHs singularity also naturally explains the non-monotonic behavior of $T_0$ as a function of doping, which is also found in mean field theories of vHs induced nematicity[46,47]. Interestingly non-Fermi liquid behavior has been argued to occur generically

of $T_0$ is on the order of 30–60 K in Fe SC[21], but is not known in the case of cuprates. Recent elasto-resistance measurements suggest that the nemato-elastic coupling might be weaker in cuprates[39–41]. Irrespective of the presence of a nematic QCP, it is clear that the doping evolution of $T_0$ contrasts with the canonical behavior of a QCP where $T_0$ would evolve monotonically as a function of the tuning parameter.

near a nematic QCP if the coupling to the lattice is weak enough[39,48,49], and the observed critical nematic fluctuations may therefore play a central role in the the strange metal properties observed in this doping range. While our results suggest that the nematic instability is linked to the proximity of vHs, we stress that the observed enhancement cannot be merely a consequence of a high-density of states at the $(\pi, 0)$ points: both $B_{1g}$ and $A_{1g}$ form factors have finite weight at these points, but only the nematic $B_{1g}$ susceptibility shows fingerprints of critical behavior at $p^*$. Thus electronic interactions in the nematic channel appears to be essential to explain our observation. We note that the key role of interactions, beyond density of state effects, was also argued to explain the divergence of the electronic specific heat coefficient which was observed at the PG end-point of several one-layer cuprates[20].

What are the consequences of our findings for the nature of the PG state? The above discussion and the doping dependence of $T_0$ allow us to conclude that the PG is likely not driven by a nematic instability. If this was the case one would expect strong nematic fluctuations close to $T^*$ in the underdoped composition and a monotonic increase of $T_0$, crossing-over to positive values, as observed in the case of Fe SC[22,50,51]. It therefore appears that nematic and PG instabilities are distinct, and possibly even compete. Note however that since our analysis of $T_0$ has been restricted to temperature above $T^*$ the weakening of the nematic fluctuations below $p^*$ cannot be a simple consequence of the PG order. Intriguingly, the states at the $(\pi, 0)$ point appears to be critical for both orders: while the strength of nematic fluctuations is closely tied to the closeness to the vHs at these points, it was suggested that the PG regime is characterized by a strong decoherence at these hot-spots due to AF fluctuations that set in once the Fermi surface reaches the $(\pi, 0)$ points corresponding to the AF zone boundary. This decoherence ultimately drives the Fermi surface hole-like and induces a cross-over to a PG state at low temperature[52,53]. Our results thus indicate that both nematicity and the PG state depend critically on the Fermi surface topology in the case of Bi2212. Further measurements on cuprates compounds where the PG endpoint and the change in the Fermi surface topology occur at distinct doping levels, like $Tl_2Ba_2CuO_{6+\delta}$[54,55] and $YBa_2Cu_3O_{6+\delta}$[56,57], are needed in order to clarify the nature of this link and confirm the connection between the nematic instability and the presence of a vHs.

## Methods

**Samples**. The Bi-2212 single crystals were grown by using a floating zone method. First optimal doped samples with $T_c = 90$ K were grown at a velocity of 0.2 mm per hour in air. In order to get overdoped samples down to $T_c = 65$ K, the as-grown single crystal was put into a high oxygen pressured cell between 1000 and 2000 bars and then was annealed from 350 °C to 500 °C during 3 days. The overdoped samples below $T_c = 60$ K were obtained from as-grown Bi-2212 single crystals put into a pressure cell (Autoclave France) with 100 bars oxygen pressure and annealed from 9 to 12 days at 350 °C. Then the samples were rapidly cooled down to room temperature by maintaining a pressure of 100 bars. The underdoped sample was obtained by annealing the as-grown sample in vacuum. The critical temperature $T_c$ for each crystal has been determined from magnetization susceptibility measurements at a 10 Gauss field parallel to the c-axis of the crystal. The selected crystals exhibit a quality factor of $\frac{T_c}{\Delta T_c}$ larger than 7. $\Delta T_c$ is the full width of the superconducting transition. A complementary estimate of $T_c$ was achieved from electronic Raman scattering measurements by defining the temperature from which the $B_{1g}$ superconducting pair breaking peak collapses. Special care has been devoted to select single crystals which exhibit the same SC pair-breaking peak energy in the Raman spectra measured from distinct laser spots on a freshly cleaved surface. The level of doping p was defined from $T_c$ using Presland and Tallons equation[58]:

$$1 - \frac{T}{T_{c,max}} = 82.6 \times (p - 0.16)^2 \qquad (3)$$

**Details of the Raman spectroscopy experiments**. Raman experiments have been carried out using a triple grating JY-T64000 spectrometer in subtractive mode

using two 1800 grooves/mm gratings in the pre-monochromator stage and 600 grooves/mm or 1800 groove/mm grating in the spectrograph stage. The 600 grooves/mm grating was used for all measurements except those carried on the OD80 sample, for which a 1800 grooves/mm grating was used. For several crystals, both configurations were used at selected temperatures to check for consistency. The 600 grooves/mm configuration allows us to cover the low-energy part of the spectrum down to $50\,cm^{-1}$ and up to $900\,cm^{-1}$ in a single frame. With the 1800 grooves/mm grating, measurements could be performed down to $15\,cm^{-1}$, but spectra must then be obtained in two frames. The resolution is set at $5\,cm^{-1}$ when using the 600 grooves/mm configuration. The spectrometer is equipped with a nitrogen cooled back illuminated CCD detector. We use the 532 nm excitation line from a diode pump solid state laser. Measurements between 10 and 300 K have been performed using an ARS closed-cycle He cryostat.

All the raw spectra have been corrected for the Bose factor and the instrumental spectral response. They are thus proportional to the imaginary part of the Raman response function $\chi''(\omega, T)$. A potential concern when correcting the raw spectra with the Bose factor is the potential presence of non-Raman signal in the raw spectra. To assess this potential non-Raman signal we note that a single effective spot temperature was able to reproduce the measured Stokes spectrum from the anti-Stokes spectrum between 20 and $600\,cm^{-1}$ at room temperature. In addition the raw Raman spectra were found to extrapolate very close to zero at zero Raman shift at the lowest temperatures measured. Both facts indicates negligible non-Raman background in the measured spectra.

The direction of incoming and outgoing electric fields are contained in the $(ab)$ plane. The $A_{1g} + B_{2g}$ geometries are obtained from parallel polarizations at 45° from the Cu–O bond directions; the $B_{2g}$ and $B_{1g}$ geometries are obtained from crossed polarizations along and at 45° from the Cu–O bond directions, respectively. The crystal was rotated in-situ using an Attocube piezo-rotator to align the electric field with respect to the crystallographic axis. $A_{1g}$ spectra are obtained from the previously listed geometries (see Supplementary Note 1 and Supplementary Fig. 1).

## Data availability

All data generated or analyzed during this study are included in the published manuscript and the supplementary information files. The relevant raw data file are available at the following url: https://doi.org/10.6084/m9.figshare.9906275.v1

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

## Acknowledgements

We thank T. Shibauchi for fruitful discussions. The work at Brookhaven National Laboratory was supported by the Office of Science, U.S. Department of Energy under Contract No. DE-SC0012704.

## Author contributions

N.A., B.L., and S.B. performed the Raman scattering experiments with the help of M.C., A.S., and Y.G. N.A. performed the data analysis and prepared the figures. R.D.Z., J.S., G. Gu. grew the single crystals and the annealing procedure to obtain underdoped and overdoped compositions. A.F. and D.C. performed the high-pressure annealing of the crystals for the strongly overdoped compositions. I.P. provided theoretical insights. A.S. and Y.G. supervised the project. Y.G. wrote the paper with inputs from all the authors.

## Competing interests

The authors declare no competing interests
