## [Peer Review File · Nature Communications]

Reviewers' comments:

Reviewer #1 (Remarks to the Author):

The authors report an extensive study of the Raman response in the cuprate superconductor Bi2212, measured at temperatures above the superconducting transition temperature T_c , up to 300 K, in 6 single crystals with different dopings, ranging from one underdoped sample with $T_c = 85$ K to a strongly overdoped sample with $T_c = 52$ K.

This study extends the previous study by the same team published in 2015 [Benhabib et al., PRL (2015)]. In that earlier study, they showed that the magnitude of the normal-state Raman B1g response in Bi2212 peaks at $p = 0.22$ ($T_c = 60$ K), and they attributed this maximum to the Lifshitz transition associated with the van Hove singularity in the band structure (where the Fermi surface goes from electron-like at high doping to hole-like at low doping). In the present study, this same enhancement of the Raman B1g response is clearly seen in Fig. 1c, where the low-energy response is clearly largest in the OD60 sample.

In the present paper, the authors ask the following question: is there also a signature of quantum criticality in the Raman response, beyond the band structure evolution through its van Hove point?

Their approach is to look for nematicity in the same way that Gallais and co-workers have successfully done it in the iron-based superconductor Ba(Fe,Co)2As2 [Gallais PRL 2013]. In that material, they did find a clear signature of nematicity and of a quantum critical point: the B1g response is characterized by a Curie-Weiss temperature T_0 that goes to zero at a critical doping x^* , being positive below x^* and negative above x^* .

In the current manuscript, we see that the same analysis of the Bi2212 data yields values of T_0 that never go to zero. Instead T_0 remains negative and large at all dopings, albeit with some variation in p (with a minimum at $p = 0.22$). In my opinion, these data show that there is no nematic quantum criticality in Bi2212 – the opposite of the conclusion drawn by the authors in their title and text.

If the pseudogap phase which sets in below T^* and ends at $p^* = 0.22$ in Bi2212 were a nematic phase, as claimed for example in ref. 18 [Sato et al., Nat. Phys. (2017)] for YBCO, then the Raman analysis should have yielded a change of sign in T_0 across p^* , with T_0 vanishing at p^* . This is not at all what is seen in the data.

I would therefore recommend that the authors re-consider entirely the conclusions they draw from their study. What they have in fact done, in my opinion, is disprove the notion that the pseudogap phase in cuprates is a nematic phase. This is a very important result. It would help if they summarized at the outset what was actually observed in their nice case study on Ba(Fe,Co)2As2, and contrast this with the behaviour found in Bi2212.

As to why they see some (non-quantum-critical) variation in T_0 , they should examine further whether that could simply come from the band structure evolution.

The paper cannot be published in its current state, for its conclusions are in my view misleading.

Minor comments:

1) In the abstract, the authors say "We further show that the nematic instability weakens upon entering the pseudogap regime, suggesting a non-trivial link between the pseudogap phenomenon and quantum criticality". This is not quite correct, for the data they use to characterize nematicity are

mostly those above T^* , and therefore outside the pseudogap phase. In other words, the drop in T_0 vs p below p^* in the lower panel of Fig. 3 is not due to the pseudogap phase.

2) They should update some of the references: 11, 19, 30, 38.

Reviewer #2 (Remarks to the Author):

The manuscript reports Raman spectroscopy measurements on a series of single crystalline samples from the Bi2212 cuprate family spanning the region of the phase diagram where, according to other experimental findings, the normal state pseudogap closes. By making use of the selection rules associated with the Raman response, the authors are able to show an enhancement in the Raman susceptibility in the B_{1g} channel that they interpret as evidence for a nematic quantum critical point (QCP).

The origin of the normal state pseudogap in cuprates is a longstanding mystery, though its end point has long been associated by researchers in the field with a quantum critical point of magnetic, charge or orbital order. Despite almost three decades of investigation, however, its fundamental nature remains controversial and within this context, the current manuscript is an important contribution. The data are of unquestionable quality, and I appreciate that the authors have been careful to distinguish their putative nematic QCP from the end point of the pseudogap, suggesting that the two are not necessarily one and the same thing. I am therefore minded to recommend publication of the manuscript in Nature Communications provided that the authors address the following concerns/comments:

(i) At the bottom of page 2, the authors begin to argue that two observations – the breaking of C₄ rotational symmetry in YBCO near T^* , as detected by torque magnetometry, and the enhanced electronic specific heat coefficient observed near p^* in Nd-LSCO – “put a hitherto undetected nematic QCP as a possible candidate to explain ... the pseudogap, superconductivity and the strange metal phases of the cuprates”. This is far too great a claim to make on the basis of two measurements on different cuprate families –there is indeed no experimental proof that the two observations are even related. I recommend that the authors either remove this statement or rephrase it with something more rooted in experimental fact.

(ii) Halfway down page 3, the authors state “In the context of cuprates the nematic order is expected to be along Cu-O-Cu bonds,”. Why is this necessarily the case? This statement needs a qualifying explanation for the general reader.

(iii) Also on page 3, the authors claim that the PG in Bi2212 closes around $p^* = 0.22$. They only provide one reference (by many of the current co-authors in fact) to support this. Given the controversy surrounding the location of p^* in different cuprate families, I would expect to see a few more references to support this conjecture.

(iv) On page 5, the authors perform a Kramers-Kronig transformation of the measured Raman response to obtain the relevant static susceptibility. They argue that the integration carried out to obtain this has a high-energy cut-off of 800 cm^{-1} . To my understanding, this is not a high-energy cut-off. Normally KK transformation involves integrations over all energies. The authors need to demonstrate that their choice of “high-energy cut-off” is robust and that varying λ does not impact on any of their conclusions.

(v) At the top of page 7, the authors state that "Upon further reducing doping, instead of crossing-over to positive temperatures T_0 reverses its behavior and decreases strongly indicating a significant weakening of the nematic instability below p^* where PG behavior sets-in." The authors need to explain how they can rule out simply the effect of the pseudogap here? I may have missed something here, but I am not convinced that the evolution of T_0 is solely an indication of the weakening of the nematic instability.

(vi) At the bottom of page 7, the authors argue that the nematic QCP occurs at the point $p^* = 0.22$ where the Fermi level crosses the van Hove singularity and the Fermi surface changes from hole-like to electron-like. The authors need to state here that this occurs only for one of the two Fermi surface sheets of the bilayer Bi2212. The crossing for the second sheet is expected to occur at a significantly higher p value.

(vii) In the last sentence of the main manuscript (page 9), the authors state that "Our results indicate that the driving force behind the PG and the QCP observed in the overdoped regime are not directly related, but both appear to depend critically on the Fermi surface topology." The authors need to comment here on the universality of this result for the cuprates in general. I think this is an important point that needs to be stressed.

Reviewer #3 (Remarks to the Author):

This manuscript reports on an interesting Raman Scattering study of the Bi2212 family of cuprate superconductors near and beyond optimum dopings, and on either side of a putative quantum critical point.

The authors report a strong temperature dependence to the Raman scattering in the B1g polarization, centred on compositions near $p^* \sim 0.22$ as exhibited in sample OD60. They make the case that this enhanced temperature dependence is due to enhanced fluctuations in the nematicity of the system, as may arise due to Pomeranchuk instabilities. This then serves as significant evidence for fluctuations associated with a quantum critical point associated with the end of the pseudo gap phase, and within the high T_c superconducting dome. This then tends to support recent thermodynamic evidence for such a quantum critical point as a generic feature of hole-doped cuprate superconductors.

I feel this is an interesting result, but have three technical comments which should be addressed before publication. I feel the first of these is quite important.

1- The description of the samples is inadequate. The samples are described as single crystals of $\text{Bi}_2\text{Ca}_2\text{SrCu}_2\text{O}_{(8+\delta)}$, but their location on a general phase diagram is indicated with a value of p , where p^* the quantum critical point is ~ 0.22 in this system. However, how is p related to the chemical composition? Is this a proxy for δ , or is the doping controlled by some other impurity? If so, what is that impurity? The central point is: what are the chemical compositions of the 6 samples, and which feature of the composition controls the doping? Also, how is this stoichiometry quantified? How were the samples characterized, beyond through the reference to their superconducting T_c s. A proper description of the samples must be provided to take this experimental study seriously.

2- The authors refer to isolating "X" from the Raman signal, which involves isolating the true signal from background and then correcting by the Bose factor. However it is important to only correct the true signal for the Bose factor - not to apply the Bose factor correction to the background. The authors

should include enough information regarding this protocol so that the readership can appreciate that this can be done cleanly with the quality of the data in the study. If necessary this could be put in the supplemental information.

2- The authors refer to "Fe SC" in the manuscript without reference to what this means. I can guess that this is iron-based superconductors, but this should be spelled out.

Reviewer #1 (Remarks to the Author):

“In the current manuscript, we see that the same analysis of the Bi2212 data yields values of T_0 that never go to zero. Instead T_0 remains negative and large at all dopings, albeit with some variation in p (with a minimum at $p = 0.22$). In my opinion, these data show that there is no nematic quantum criticality in Bi2212 – the opposite of the conclusion drawn by the authors in their title and text.

If the pseudogap phase which sets in below T^* and ends at $p^* = 0.22$ in Bi2212 were a nematic phase, as claimed for example in ref. 18 [Sato et al., Nat. Phys. (2017)] for YBCO, then the Raman analysis should have yielded a change of sign in T_0 across p^* , with T_0 vanishing at p^* . This is not at all what is seen in the data.

I would therefore recommend that the authors re-consider entirely the conclusions they draw from their study. What they have in fact done, in my opinion, is disprove the notion that the pseudogap phase in cuprates is a nematic phase. This is a very important result. It would help if they summarized at the outset what was actually observed in their nice case study on $\text{Ba}(\text{Fe},\text{Co})_2\text{As}_2$, and contrast this with the behaviour found in Bi2212.”

Reply: We broadly agree with the referee. Our data indeed demonstrate that the PG is unlikely primarily driven by nematic degrees of freedom. At best the nematic order observed by Sato et al. is likely to be a secondary instability. This is in fact what we already stated in the discussion “... allow us to conclude that the PG is likely not driven by a nematic instability”. However, we understand from the referee that this important conclusion was not sufficiently highlighted in the abstract of the ms. As to whether Nematic Quantum criticality occurs in Bi2212 or not, we would not be as definitive as the referee. First T_0 values are strongly doping dependent near $p=0.22$: above 0.22 they interpolate to 0 very close to $p=0.22$, at a doping above the next data point which correspond to OD74. Therefore, our data cannot rule out a positive T_0 value between OD60 and OD74 compositions. Second the extracted T_0 Curie-Weiss temperatures do not include the contribution from the lattice which, as shown in Fe SC, will push the actual thermodynamic nematic transition temperature (in the absence of SC) to higher values. This electron-lattice coupling energy scale is currently not known in the cuprates. For these reasons we feel a nematic quantum criticality cannot be ruled out. On the other hand, if there is indeed a nematic ground state around $p=0.22$, there must be two QCPs because, clearly, well-above and well-below this doping the nematic correlation length is only weakly diverging. Either that, or around $p=0.22$ there is a missed nematic QCP such that the nematic correlations are enhanced only around this special doping for reasons that need to be fully understood in future studies, but that are likely related to the presence of a van Hove singularity.

Following the suggestions of the referee we have modified both the title and the abstract of the ms in order to more faithfully represent the main conclusions of our work, in particular with

respect to the link between PG and nematicity. We have also modified the discussion which now includes an assessment about the existence of nematic quantum criticality or not in Bi2212.

“As to why they see some (non-quantum-critical) variation in T_0 , they should examine further whether that could simply come from the band structure evolution.”

The connection to FS topology was made in our discussion when we stated that the nematic instability is linked to the proximity to a van Hove singularity. We agree with the referee that such connection naturally explains the non-monotonic behavior of T_0 with doping since T_0 decreases when the Fermi level moves away from the vHs. This is also what is found in mean-field theories of vHs induced nematicity (Khavkine et al. PRB 2004 and Yamase et al. PRB 2005, these 2 references were added in the ms). Note however that our results cannot be a simple band structure effect as the A1g response, which also probes the van Hove points, does not display this non-monotonic behavior. We therefore believe that interactions in the nematic channel, whose origin remains to be clarified, must play a role too. We have modified the discussion to include this aspect. In particular we now state that the proximity to a vHs naturally explains the doping dependence of T_0 .

“The paper cannot be published in its current state, for its conclusions are in my view misleading.”

We believe our modified version now better highlights our conclusions about the link between PG and nematicity. It also includes a more critical discussion of the presence or not of a nematic QCP.

“Minor comments:

1) In the abstract, the authors say “We further show that the nematic instability weakens upon entering the pseudogap regime, suggesting a non-trivial link between the pseudogap phenomenon and quantum criticality”. This is not quite correct, for the data they use to characterize nematicity are mostly those above T^* , and therefore outside the pseudogap phase. In other words, the drop in T_0 vs p below p^* in the lower panel of Fig. 3 is not due to the pseudogap phase.”

We agree. By PG regime we meant doping level where PG is observed. This point is discussed in the last part of the discussion “Since our analysis of T_0 has been restricted to temperature above T^* ... sub-leading”. We understand that the formulation “PG regime” in the abstract might be misleading and we have removed it from the abstract. We have also added a sentence in the discussion to better clarify this important point.

“2) They should update some of the references: 11, 19, 30 and 38”

We have updated ref. 19 (Michon et al.) and 38 (Oganesyan et al.). Ref. 11 and 30 are not yet published to our knowledge.

Reviewer #2 (Remarks to the Author):

“(i) At the bottom of page 2, the authors begin to argue that two observations – the breaking of C4 rotational symmetry in YBCO near T^* , as detected by torque magnetometry, and the

enhanced electronic specific heat coefficient observed near p^* in Nd-LSCO – “put a hitherto undetected nematic QCP as a possible candidate to explain ... the pseudogap, superconductivity and the strange metal phases of the cuprates”. This is far too great a claim to make on the basis of two measurements on different cuprate families –there is indeed no experimental proof that the two observations are even related. I recommend that the authors either remove this statement or rephrase it with something more rooted in experimental fact.”

We have rephrased this sentence in order to take a more neutral stance, stressing that the existence and possible role of a nematic QCP remain to be addressed.

“(ii) Halfway down page 3, the authors state “In the context of cuprates the nematic order is expected to be along Cu-O-Cu bonds,”. Why is this necessarily the case? This statement needs a qualifying explanation for the general reader. “

This sentence was motivated by the fact that most of experimental works have reported anisotropies along the Cu-O-Cu direction (ref. 12, 13, 14, 15, 16, 18). Similarly most theoretical works on cuprates indicate a predominant instability along this direction (see for e. g. ref. 35, 36). However, we agree that this may not be necessarily the case and may depend on both systems and doping regime. In fact the recent work of Murayama et al. on Hg-1201 (arxiv 2018, unpublished) and Wu et al. on LSCO (Nature , 2017) have also revealed nematicity along different directions than the Cu-O-Cu bonds. We have modified this sentence to reflect the more complex state of affair, and added references to Murayama et al. and Wu et al..

“(iii) Also on page 3, the authors claim that the PG in Bi2212 closes around $p^* = 0.22$. They only provide one reference (by many of the current co-authors in fact) to support this. Given the controversy surrounding the location of p^* in different cuprate families, I would expect to see a few more references to support this conjecture. “

We agree that this value of p^* is not universally accepted and is likely to be material dependent (for e.g. $p^*=0.19$ in YBCO). In Bi2212 the evolution of PG inside the SC region is not yet settled with some reports claiming back-bending behavior of T^* below T_c . However, focusing on the behavior above T_c , there is a relatively wide consensus on Bi2212 that no PG is observed beyond $p=0.22$ while finite T^* is observed below. Since our work is focused on the behavior above T_c we believe that this value of p^* is the most relevant, as the $T=0$ critical doping might be affected by phase competition between SC and PG. We have added new references that substantiate our statement about the doping dependence of T^* in Bi2212: ref. 27 (Vishik et al.) and new references: Ishida et al. PRB 1997, Oda et al. Physica C 1997,, Dipasupil et al. J. Phys. Soc. Jp 2002), Usui et al. J. Phys. Soc. Jap 2014. and Hashimoto et al. (Nature Mater. 2015).

“(iv) On page 5, the authors perform a Kramers-Kronig transformation of the measured Raman response to obtain the relevant static susceptibility. They argue that the integration carried out to obtain this has a high-energy cut-off of 800 cm^{-1} . To my understanding, this is not a high-energy cut-off. Normally KK transformation involves integrations over all energies. The authors need to demonstrate that their choice of “high-energy cut-off” is robust and that varying λ does not impact on any of their conclusions.”

The cut-off was chosen as the energy scale at which the spectra are temperature independent for all doping. A close inspection of the data reveals that this energy scale is about 800 cm⁻¹ for the doping range studied. Integrating further in energy will therefore only add a temperature independent constant to the static susceptibility. Since we are only interested in the temperature dependence of the susceptibility, extending further in energy will not affect the T dependent behavior of χ . Furthermore, because the quantity integrated is χ/ω and not χ the contribution to the susceptibility coming from the high energy range is small, and only weakly affect the estimation of T_0 . This is shown in the figure below where Curie-Weiss fit (ignoring the constant) and T_0 values have been extracted using different cut-off ranging from 700 to 980 cm⁻¹. While T_0 values can change by as much as 20%, this remains within the error bar of the fits for most doping. More importantly the overall behavior of T_0 with doping is identical and is therefore robust with the choice of cut-off. The main conclusions of our study are therefore not affected by the choice of cut-off. We have added a figure in the SM showing the T_0 values for different cut-off as a function of doping.

“(v) At the top of page 7, the authors state that “Upon further reducing doping, instead of crossing-over to positive temperatures T_0 reverses its behavior and decreases strongly indicating a significant weakening of the nematic instability below p^* where PG behavior sets-in.” The authors need to explain how they can rule out simply the effect of the pseudogap here? I may have missed something here, but I am not convinced that the evolution of T_0 is solely an indication of the weakening of the nematic instability.”

As discussed at the end of the discussion, and as pointed by referee #1, since the Curie-Weiss fit are performed above T^* the PG order cannot explain the weakening of the nematic correlations. The confusion arises because of the use of “PG regime” terminology in the previous version of the manuscript. We have added a sentence in the discussion to stress this important point.

“(vi) At the bottom of page 7, the authors argue that the nematic QCP occurs at the point $p^* = 0.22$ where the Fermi level crosses the van Hove singularity and the Fermi surface changes from hole-like to electron-like. The authors need to state here that this occurs only for one of the two Fermi surface sheets of the bilayer Bi2212. The crossing for the second sheet is expected to occur at a significantly higher p value.”

We have added a sentence in the text which defines more accurately the change in FS topology in the case of a bi-layer cuprate like Bi2212.

“(vii) In the last sentence of the main manuscript (page 9), the authors state that “Our results indicate that the driving force behind the PG and the QCP observed in the overdoped regime are not directly related, but both appear to depend critically on the Fermi surface topology.” The authors need to comment here on the universality of this result for the cuprates in general. I think this is an important point that needs to be stressed.”

This is a good point. While the end of PG appears to correspond to a change in FS topology in several cuprates (like Bi2212, Bi2201, Nd-LSCO), this is not universally the case. For example, in YBCO and Tl2201 they occur at 2 different doping levels. Rather the rule seems that PG only sets-in on hole-like Fermi-surface: they are no example to our knowledge of a PG being observed on an electron-like Fermi surface. This apparent rule has been discussed

theoretically in for e.g. ref. 51 and 52 (Wu et al. Braganca et al.). Another important point which need sto be addressed is whether the enhancement of nematic fluctuations is connected to the presence of a van-Hove singularity as we propose in our study in Bi2212, or if it is connected to the end-point of the PG. while we believe the former scenario is more likely, they can only be distinguished by studying a cuprate system where the two dopings are separated (like YBCO or Tl2201). We now include a discussion on this aspect at the very end of the paper.

Reviewer #3 (Remarks to the Author):

“1- The description of the samples is inadequate. The samples are described as single crystals of $\text{Bi}_2\text{Ca}_2\text{SrCu}_2\text{O}_{(8+\delta)}$, but their location on a general phase diagram is indicated with a value of p , where p^* the quantum critical point is ~ 0.22 in this system. However, how is p related to the chemical composition? Is this a proxy for δ , or is the doping controlled by some other impurity? If so, what is that impurity? The central point is: what are the chemical compositions of the 6 samples, and which feature of the composition controls the doping? Also, how is this stoichiometry quantified? How were the sample characterized, beyond through the reference to their superconducting T_c s. A proper description of the samples must be provided to take this experimental study seriously.”

All our sample come originally from the same batch of nearly optimally doped Bi2212 single crystals with a T_c of 90K. The doping of these as-grown crystals was tuned only using oxygen content δ . The as-grown crystals were then annealed under high oxygen pressures in order to obtain overdoped compositions (see SI of ref. 28 Benhabib et al.). Annealing of the crystal under vacuum was performed in order to reach the underdoped composition studied here. Their T_c was checked using a SQUID magnetometer. Only crystals with sharp superconducting transitions were selected. Since actual oxygen stoichiometry is hard to quantify and thus absolute doping levels, we rely on both the T_c and the superconducting gap energy as proxies for their relative doping level. The doping was deduced using the phenomenological Presland-Tallon law linking T_c , $T_{c,\text{max}}$ and the hole doping p . T_c and sample homogeneity were also checked by Raman via the superconducting gap energy (which scale with T_c in the overdoped composition) and the sharpness of the superconducting pair-breaking peak (see Ref. 28 Benhabib et al.). In addition, ARPES measurements on two overdoped Bi2212 single crystals coming from the same batch were performed recently (ref. 35 Loret et al. PRB 87, 174521 (2018)). The doping could be evaluated from the FS area using Luttinger theorem and shown to be in good agreement with the one deduced from Presland-Tallon law. All these details are now provided in the SM section.

“2- The authors refer to isolating X from the Raman signal, which involves isolating the true signal from background and then correcting by the Bose factor. However it is important to only correct the true signal for the Bose factor - not to apply the Bose factor correction to the background. The authors should include enough information regarding this protocol so that the readership can appreciate that this can be done cleanly with the quality of the data in the study. If necessary this could be put in the supplemental information.”

In general the collected intensity can contain both Raman and non-Raman like (for e.g. fluorescence) signal. The latter typically arises from impurities and parasitic phases at the sample surface and potentially from the optical set-up itself. In the case of Bi2212 crystals, the quality of the cleaved surface is such that the non-Raman signal below 600 cm^{-1} was found to be essentially negligible. This was evaluated using 2 methods. First both Stokes and Anti-Stokes spectra were collected at elevated temperature ($>200\text{K}$) and excellent agreement was found from 30 to 600 cm^{-1} . Any significant non-Raman background would yield inconsistent Stokes and anti-Stokes spectra. Second the raw spectra (not Bose corrected) at the lowest temperature (10K in our case) should extrapolate to zero when considering the data above 25 cm^{-1} . Since at 10K the Bose factor is essentially one above 25 cm^{-1} , any non-zero extrapolation would indicate non-Raman signal since the dissipative part of the Raman response function is an odd function. In our case the low temperature signal extrapolates very close to zero, and the small finite positive intercept is essentially within our experimental accuracy, and was found to have a negligible impact on the value of χ and the overall analysis. Example of these 2 methods are shown in the figures below on an as-grown single and an OD74 crystal.

We have added these details to the supplementary material section.

“2- The authors refer to "Fe SC" in the manuscript without reference to what this means. I can guess that this is iron-based superconductors, but this should be spelled out.”

This was corrected in the revised version.

REVIEWERS' COMMENTS:

Reviewer #1 (Remarks to the Author):

I am satisfied with the changes made by the authors.

The text is now much more open about the possible connection between their nematic fluctuations and the pseudogap phase.

The abstract now gives a more balanced account of the likely relation between nematicity and pseudogap and van Hove singularity.

My only recommended change is this:

Given the changes in emphasis, it now seems odd to highlight the connection to the pseudogap phase in the title.

I would recommend the title be changed to simply:

"Nematic fluctuations in the cuprate superconductor Bi22121"

For indeed, the clear and major contribution of this work is to have, for the first time as far as I know, measured the nematic FLUCTUATIONS in a cuprate superconductor.

In this sense, this is an important contribution to the field.

Much like the earlier Raman work in iron-based superconductors, but even more so since in cuprates we do not have the remarkable least-resistivity studies reported for iron-based superconductors.

Reviewer #2 (Remarks to the Author):

I am satisfied with the response of the authors to the referee comments. The manuscript is markedly improved (and less confusing!) as a result and I am happy to recommend its publication in Nature Communications.

Reviewer #3 (Remarks to the Author):

The authors have satisfactorily responded to my three comments. I have also read through the comments and responses associated with the two other referees - these also seem like adequate responses, so I would recommend acceptance at this point.